# EBI2-mediated bridging channel positioning supports splenic dendritic cell homeostasis and particulate antigen capture

Tangsheng Yi[1,2], Jason G Cyster[1,2]*

[1]Department of Microbiology and Immunology, University of California, San Francisco, San Francisco, United States; [2]Howard Hughes Medical Institute, University of California, San Francisco, San Francisco, United States

**Abstract** Splenic dendritic cells (DCs) present blood-borne antigens to lymphocytes to promote T cell and antibody responses. The cues involved in positioning DCs in areas of antigen exposure in the spleen are undefined. Here we show that CD4[+] DCs highly express EBI2 and migrate to its oxysterol ligand, 7α,25-OHC. In mice lacking EBI2 or the enzymes needed for generating normal distributions of 7α,25-OHC, CD4[+] DCs are reduced in frequency and the remaining cells fail to situate in marginal zone bridging channels. The CD4[+] DC deficiency can be rescued by LTβR agonism. EBI2-mediated positioning in bridging channels promotes DC encounter with blood-borne particulate antigen. Upon exposure to antigen, CD4[+] DCs move rapidly to the T-B zone interface and promote induction of helper T cell and antibody responses. These findings establish an essential role for EBI2 in CD4[+] DC positioning and homeostasis and in facilitating capture and presentation of blood-borne particulate antigens.

*For correspondence: jason.cyster@ucsf.edu

Competing interests: The authors declare that no competing interests exist.

## Introduction

Dendritic cells (DCs) play a crucial role in presenting antigens to T cells to initiate adaptive immune responses (*Banchereau and Steinman, 1998*). The DC lineage is divided into several subtypes based on transcription factor requirements, surface markers and ability to prime CD4 vs CD8 T cell responses. By surface phenotype, the spleen contains three major populations of conventional DCs: CD4[+], CD8[+] and double negative (DN) DCs (*Shortman and Liu, 2002*; *Hashimoto et al., 2011*; *Miller et al., 2012*). CD4[+] DCs also express the C-type lectin receptor called DC-inhibitory receptor 2 (DCIR2), identified by the antibody 33D1, whereas CD8[+] DCs express the lectin DEC205 (*Dudziak et al., 2007*). Lymph nodes (LNs) contain these DC populations but also contain additional subsets that travel to these organs from peripheral tissues sites (*Hashimoto et al., 2011*). Based on soluble antigen immunization studies and antigen targeting to DC subtypes using antibodies, as well as studies in mice deficient in DC subsets, CD4[+] and DN DCs have been most strongly implicated in priming CD4 T cell responses against exogenous antigens whereas CD8[+] DCs have a more prominent role in cross-presenting antigens and promoting CD8 T cell responses (*Pooley et al., 2001*; *Dudziak et al., 2007*; *Hildner et al., 2008*). However, these functional distinctions are not strict with some studies, for example, suggesting that CD8[+] DCs contribute to priming of CD4[+] T cell responses (*Tamura et al., 2005*; *Fukaya et al., 2012*).

The spleen, the largest secondary lymphoid organ, has an essential function in supporting rapid B and T cell responses against circulating antigens (*Mebius and Kraal, 2005*). Blood entering the spleen is released at the marginal sinus prior to travelling through the marginal zone in a direction away from the lymphoid (white pulp) regions, into the red pulp to return to circulation via venous sinuses. The

**eLife digest** One of the main roles of the spleen is to make the antibodies that protect the body against viruses, bacteria and other microorganisms. Antibodies are made by B cells, which are a type of white blood cell, after they have been exposed to antigens. For most antibody responses, it is also necessary for the B cells to get help from other white blood cells called T cells that have been exposed to antigens. Specialized cells called dendritic cells have a central role in bringing the antigens—which are usually fragments of the infectious agents that have invaded the body—to the T cells.

One subset of dendritic cells, called CD4+ dendritic cells, are found in large numbers in a part of the spleen called the bridging channel, but the process by which these cells become localized in this channel has not been fully understood. Now, Yi and Cyster show that a receptor called EBI2, which is found on the surface of CD4+ dendritic cells, binds to a type of organic molecule called an oxysterol that is produced in the bridging channel.

In mice that had been genetically engineered to lack EBI2 or the enzymes needed to make this particular oxysterol—which is known as 7α,25-dihydroxycholesterol, or 7α,25-OHC for short—the CD4+ dendritic cells were no longer clustered in the bridging channel and their number was markedly decreased. This showed that the interaction between EBI2 and the oxysterol was essential for ensuring that the CD4+ dendritic cells were in the right place. The correct positioning of the CD4+ dendritic cells was, in turn, necessary for maintaining cell numbers. Moreover, these mice had a weakened immune response because of the very low number of antigens that were being presented to the T cells.

A number of autoimmune diseases, such as lupus, are caused by the body developing an immune response to its own cells and tissues. One implication of the work of Yi and Cyster is that if small molecule inhibitors of EBI2 could be designed, they might be able to suppress the onset of such autoimmune responses.

CD4+ 33D1+ DCs in the spleen are enriched in an area of the white pulp known as the marginal zone (MZ) bridging channel, where the T zone abuts the blood-rich red pulp (*Mitchell, 1973*; *Witmer and Steinman, 1984*; *Steinman et al., 1997*). CD8+ DEC205+ DCs, by contrast, are mostly located within the T zone (also known as the periarteriolar lymphatic sheath or PALS). The factors that promote CD4+ DC positioning in MZ bridging channels have been unclear.

The developmental requirements of DCs have been under intense investigation. DCs derive from pre-DCs in the bone marrow (BM) and these cells seed the spleen and give rise to the CD4+, CD8+ and DN DC subsets (*Naik et al., 2006*; *Liu et al., 2007*). The transcription factors IRF2, IRF4, RelB and RBP-J are needed for CD4+ DC development whereas Batf3, IRF8 and Id2 are required for CD8+ DC development (*Hashimoto et al., 2011*; *Satpathy et al., 2012*). CD4+ DCs require signaling by Notch to induce RBP-J (*Lewis et al., 2011*), and by the cytokine LTα1β2, the latter coming from B cells (*Kabashima et al., 2005*). How maturing DCs are guided into locations that ensure receipt of appropriate developmental or homeostasis signals is not understood.

EBI2 (GPR183) is a Gαi coupled receptor that responds to the oxysterol ligand, 7α,25-hydroxycholesterol (OHC) (*Hannedouche et al., 2011*; *Liu et al., 2011*). EBI2 is upregulated by activated B cells and plays an important role in promoting their movement to inter- and outer-follicular regions of lymphoid organs during T-dependent antibody responses (*Gatto et al., 2009*; *Pereira et al., 2009*). 7α,25-OHC is synthesized from cholesterol by the actions of two enzymes, CH25H and CYP7B1, and these are expressed at high levels by stromal cells in inter- and outer-follicular regions while expression of Ch25h is repressed in the center of follicles (*Yi et al., 2012*). As well as expression by B cells, EBI2 is present on a variety of other hematopoietic cell types, including DCs (*Hannedouche et al., 2011*; *Liu et al., 2011*). However, its function in DCs is unknown.

Here we show that EBI2- and EBI2-ligand deficient mice have a marked deficiency in CD4+ DCs in spleen and related DCs in LNs. CD4+ DCs migrate strongly to 7α,25-OHC and the CD4+ DCs remaining in the spleen of EBI2-deficient mice fail to localize correctly in MZ bridging channels. The CD4+ DCs in EBI2-deficient mice show evidence of reduced LTβR engagement and their numbers can be rescued by increased LTβR agonism. When situated in the MZ bridging channels, CD4+ DCs rapidly capture

particulate antigen, upregulate CCR7 and relocalize to the T-B zone interface to promote CD4 T cell and antibody responses. These responses are defective when DCs lack EBI2.

## Results

### EBI2 expression and promigratory function in DCs

CD11c$^+$ splenic DCs from EBI2-GFP reporter mice showed high GFP abundance in CD8$^-$ vs CD8$^+$ DCs (*Figure 1A*). In agreement with the pattern of reporter expression, CD4$^+$ DCs from wild-type mice had more *Ebi2* transcripts than CD8$^+$ DCs, and CD4$^+$ DCs had higher surface expression of EBI2

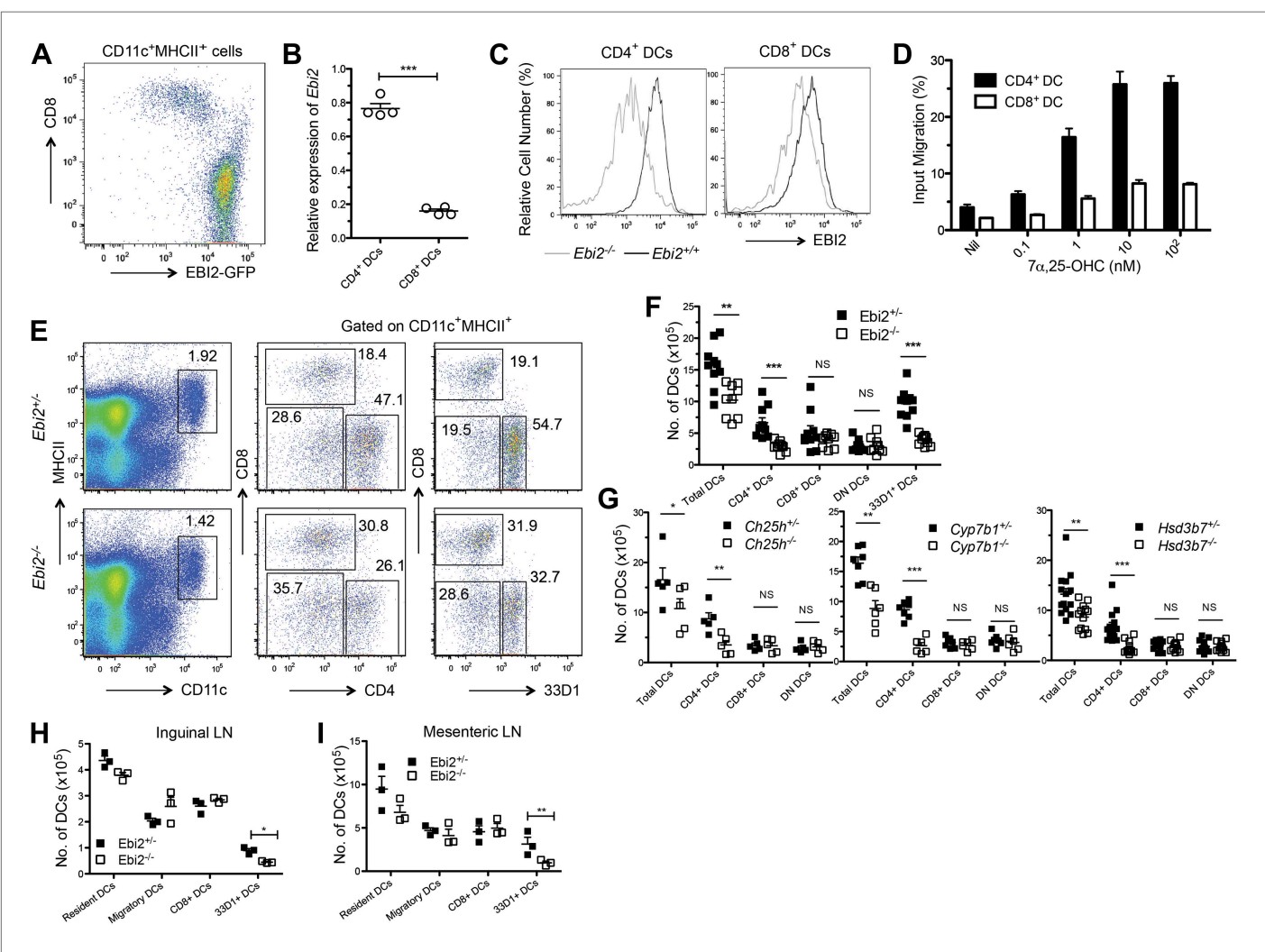

**Figure 1**. EBI2 expression in DCs and deficiency of CD4$^+$ DCs in mice lacking EBI2 or correct amounts of EBI2 ligand. (**A**) Flow cytometric detection of GFP fluorescence in gated splenic CD11c$^+$MHCII$^+$ cells from *Ebi2$^{GFP/+}$* mice. (**B**) Quantitative PCR analysis of *Ebi2* transcript abundance in sorted splenic CD11c$^+$MHCII$^+$CD4$^+$ cells (CD4$^+$ DCs) and CD11c$^+$MHCII$^+$CD8$^+$ cells (CD8$^+$ DCs). Expression is shown relative to *Hprt* (n = 4 mice). (**C**) EBI2 surface staining of gated splenic CD4$^+$ and CD8$^+$ DCs from *Ebi2$^{-/-}$* or *Ebi2$^{+/+}$* mice (one representative of four experiments). (**D**) Migration of CD4$^+$ DCs and CD8$^+$ DCs in response to 7α,25-OHC (mean from four mice ± SE, combined from two experiments). (**E**)–(**I**) Flow cytometry and enumeration of splenocytes (**E**, **F**, and **G**), inguinal LN cells (**H**), and mesenteric LN cells (**I**) from *Ebi2$^{-/-}$*, *Ch25h$^{-/-}$*, *Cyp7b1$^{-/-}$*, *Hsd3b7$^{-/-}$* mice, and their matched littermate controls. Numbers adjacent to outlined areas in **E** indicate percent cells in each gate. DN DCs are defined as CD4$^-$CD8$^-$CD11c$^+$MHCII$^+$ cells. Each dot in **F**–**I** represents an individual mouse and error bars indicate mean ± SE of samples combined from three to five independent experiments. Lymph node migratory DCs are defined as MHCII$^{hi}$CD11c$^{int}$ and resident DCs, including the CD8$^+$ and 33D1$^+$ DCs, as MHCII$^{int}$CD11c$^{hi}$. *p<0.05, **p<0.01, ***p<0.001, Student's T-test.

The following figure supplements are available for figure 1:

**Figure supplement 1**. DC properties in EBI2-deficient mice and in mice lacking CYP7B1 in radiation resistant cells.

(*Figure 1B,C*). This difference in chemoattractant receptor expression was unique to EBI2 as it was not seen for the highly expressed chemokine receptors, CCR7 and CXCR4 (*Figure 1—figure supplement 1A*). The higher EBI2 expression in CD4$^+$ DCs conferred a strong ability to chemotax in response to 7α,25-OHC in transwell assays, with the cells exhibiting migratory responses to subnanomolar concentrations of ligand (*Figure 1D*). By contrast, CD8$^+$ DCs failed to migrate to subnanomolar ligand and migration was weak even at high ligand concentrations (*Figure 1D*).

## CD4$^+$ DC deficiency in EBI2 and EBI2-ligand deficient mice

Analysis of DC subsets in EBI2-deficient mice revealed a threefold to fourfold deficiency in splenic CD4$^+$ DCs without a change in the number of CD8$^+$ DCs or DN DCs (*Figure 1E,F*). Quantitation of DCs in mice lacking either of the enzymes needed for 7α,25-OHC synthesis, CH25H or CYP7B1, showed a comparable selective loss of CD4$^+$ DCs (*Figure 1G*). Moreover, mice lacking HSD3B7, the enzyme that metabolizes 7α,25-OHC, and that have greatly increased amounts of 7α,25-OHC in lymphoid organs (*Yi et al., 2012*), had a similar deficiency of CD4$^+$ DCs (*Figure 1G*). When Cyp7b1-deficient mice were reconstituted with wild-type bone marrow, the mice remained CD4$^+$ DC deficient, indicating that radiation resistant stromal cells were a necessary source of EBI2 ligand (*Figure 1—figure supplement 1B*).

The C-type lectin DCIR2, detected with the 33D1 antibody (*Witmer and Steinman, 1984*; *Dudziak et al., 2007*), is present on all CD4$^+$ DCs and on a fraction of DN DCs (*Figure 1—figure supplement 1C*). Enumeration of 33D1$^+$ DCs showed a significant reduction of positive cells in the spleen, confirming that the reduction in CD4$^+$ DCs is due to a loss of this cell type rather than being due to a reduction in surface marker expression (*Figure 1F*). The CD4$^+$ DCs remaining in EBI2-deficient mice exhibited normal expression of the surface molecules MHC class II, CD80, CD83 and CD86 and in vitro they supported a normal mixed lymphocyte reaction (*Figure 1—figure supplement 1D,E*). Although the DC populations present in LNs are more heterogeneous than within spleen, we detected a similar reduction in 33D1$^+$ DCs in peripheral (inguinal) and mucosal (mesenteric) LNs, while CD8$^+$ DCs and migratory DCs were present at normal frequencies (*Figure 1H,I*). As in the spleen, LN 33D1$^+$ DCs expressed high amounts of EBI2 (*Figure 1—figure supplement 1F*).

To test whether EBI2 was required intrinsically in CD4$^+$ DCs we generated *Ebi2$^{-/-}$* CD45.2: WT CD45.1 mixed BM chimeras. This analysis revealed a similar reduction in CD4$^+$ DCs to that seen in fully deficient mice, establishing an intrinsic role for EBI2 in these cells and showing that the phenotype was not increased when the mutant cells had to compete with wild-type cells (*Figure 2A,B*). All other splenic DC subsets, including pDCs, were unaffected by EBI2-deficiency (*Figure 2A,B*). As another test of the intrinsic in vivo activity of EBI2 in DCs we reconstituted mice with BM cells that had been transduced with an EBI2 and hCD4-reporter expressing retrovirus or with a truncated NGFR vector control. 8 weeks post reconstitution there was a marked increase in the frequency of DCs in the spleens of mice overexpressing EBI2 and this increase was restricted to the 33D1$^+$ DC subset (*Figure 2C,D*). These data indicate that EBI2 is necessary for development or maintenance of CD4$^+$33D1$^+$ DC and elevated expression of EBI2 is sufficient to promote increased accumulation of this DC type.

## EBI2 is required for DC positioning in bridging channels

Given the strong chemoattractant activity of EBI2 ligand (*Figure 1D*) and the demonstration in our studies on B cell positioning that the enzymes required for ligand synthesis are expressed abundantly in interfollicular regions (*Yi et al., 2012*), we asked whether EBI2 deficiency led to alterations in DC positioning. In the spleen, CD4$^+$ DCs are enriched in MZ bridging channels, specialized interfollicular regions that connect the T zone with the red-pulp (*Mitchell, 1973*). Locating DCs in tissue sections by staining for CD4 is problematic due to the large abundance of CD4 T cells. We therefore took advantage of the DC restricted expression of DCIR2, detected with 33D1 (*Witmer and Steinman, 1984*; *Dudziak et al., 2007*), for this assessment. Strikingly, while 33D1$^+$ DCs were concentrated in bridging channels of control mice, they were markedly under represented in these regions in EBI2-deficient mice (*Figure 3A*). Although the total number of 33D1$^+$ DCs in the sections was reduced, consistent with the reduced cell numbers detected by flow cytometry, the 33D1$^+$ DCs that remained were mostly distributed in the red pulp or in the T zone (*Figure 3A*). The distribution of DEC205$^+$ DCs was unaltered in EBI2-deficient mice (*Figure 3B*). A similar deficit of 33D1$^+$ DCs in bridging channels was seen in *Ch25h$^{-/-}$*, *Cyp7b1$^{-/-}$* and *Hsd3b7$^{-/-}$* mice (*Figure 3A*) confirming the need for correctly distributed 7α,25-OHC to mediate 33D1$^+$ DC positioning. Although 33D1$^+$ DCs are less numerous in

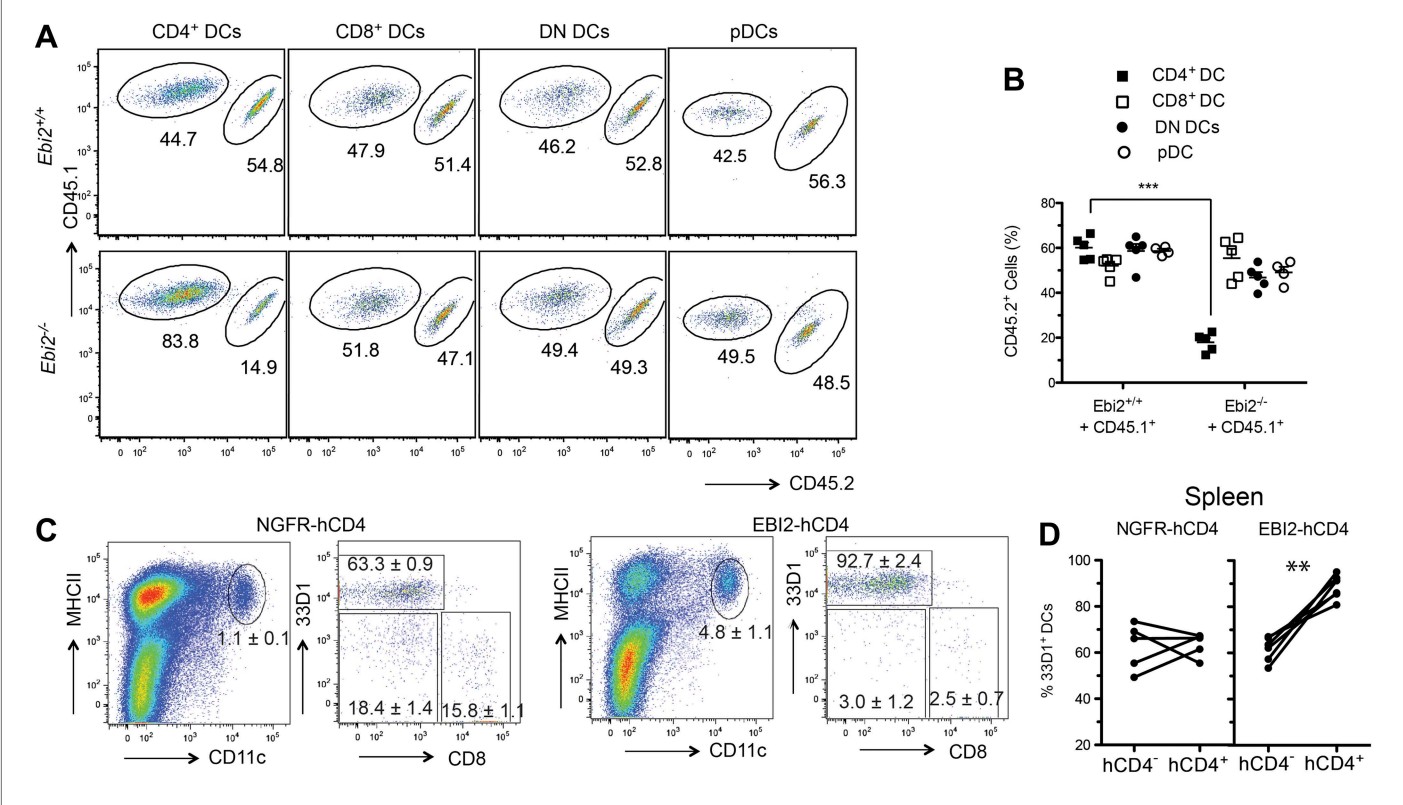

**Figure 2**. Intrinsic requirement for EBI2 in CD4⁺ DCs. (**A**) and (**B**) Wild-type CD45.2⁺ mice were lethally irradiated and reconstituted with mixed BM cells (1:1 ratio) from CD45.1⁺ *Ebi2⁺/⁺* mice and CD45.1⁺CD45.2⁺ *Ebi2⁻/⁻* mice or their wild-type littermate controls. Eight weeks after reconstitution, splenocytes were analyzed by flow cytometry for CD4⁺ DCs, CD8⁺ DCs, DN DCs (gated as in **Figure 1**), and pDCs (defined as CD11c⁺siglecH⁺MHCII⁺). (**A**) Representative flow cytometric data. (**B**) Summary of CD45.2⁺ cell frequencies for data of the type in (**A**) (n = 5 mice). (**C**) and (**D**) Mice were reconstituted with BM transduced with a retroviral construct encoding EBI2 or truncated neural growth factor receptor (NGFR), with an IRES–truncated human CD4 cassette as a reporter. Flow cytometric analysis of splenic DCs of chimeras six weeks after reconstitution (n = 5–7 mice). Plots in (**C**) are shown gated on hCD4⁺ cells and right plots are further gated on MHCII⁺CD11c⁺ cells. Plots in (**D**) are summary of frequency of 33D1⁺ DCs among total CD11c⁺MHCII⁺ cells that were non-transduced (hCD4⁻) and transduced (hCD4⁺). ***p<0.001, Student's T-test.

LNs, they were again situated most abundantly in areas between follicles in wild-type mice, and they were reduced in these regions in mice lacking EBI2 (**Figure 3C**).

## EBI2 is not required in pre-DCs

To examine the basis for the reduced numbers of CD4⁺ DCs in EBI2-deficient mice we first examined DC turnover rates using bromodeoxyuridine (BrdU) incorporation. These measurements showed that each subset of splenic DCs turned over at similar rates in EBI2-deficient and control mice (**Figure 4A**). Splenic DCs are short-lived (**Kamath et al., 2002**) and their viability can be prolonged by removal of the proapoptotic molecule Bim (**Chen et al., 2007**). However, when Cyp7b1-deficient mice were reconstituted with *Bim⁻/⁻* BM, despite having more total DCs, they continued to suffer a CD4⁺ DC deficiency (**Figure 4B**). Consistent with this finding, enumeration of annexin V⁺ DCs and active Caspase-3⁺ DCs in spleen cell suspensions failed to show differences in their frequency in wild-type and EBI2-deficient mice (not shown). These observations suggest that more rapid apoptosis does not account for the reduced numbers of CD4⁺ DCs in mice lacking EBI2 function.

EBI2 is expressed in pre-DCs though at somewhat lower levels than in CD4⁺ DCs (**Figure 4C**). However, EBI2-deficient mice contained normal numbers of pre-DCs in BM and spleen (**Figure 4D**). To test whether splenic pre-DC that had developed in the absence of EBI2 signaling retained normal developmental potential we took advantage of the finding that transfers of pre-DCs into congenically marked recipient animals gives rise to small but traceable numbers of CD4, CD8 and DN DCs 6 days later (**Naik et al., 2006**; **Liu et al., 2007**). When WT pre-DCs were transferred to *Cyp7b1⁻/⁻* recipients,

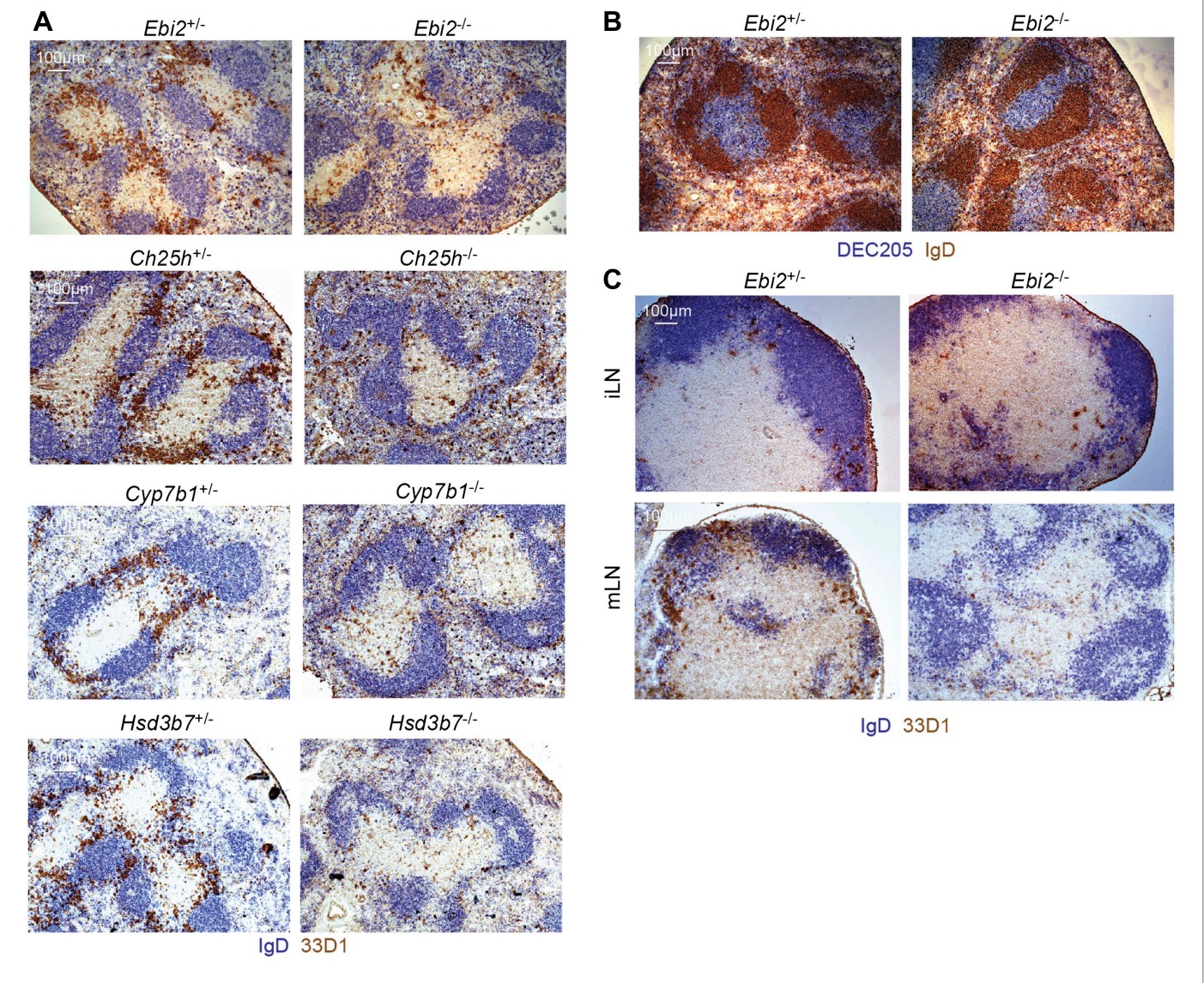

**Figure 3**. EBI2 is required for DC positioning in MZ bridging channels and LN interfollicular regions. Immunohistochemistry of spleen (**A** and **B**) and lymph node (**C**) sections from the indicated mice were stained for IgD and 33D1 or DEC205, as indicated. Data are representative of at least six mice of each type.

the number of CD4+ DCs generated was reduced, consistent with the need for EBI2 ligand to sustain these DCs (*Figure 4E*). By contrast, when *Cyp7b1⁻/⁻* pre-DCs were transferred to wild-type recipients, normal numbers of CD4+ DCs were generated (*Figure 4F*). These data demonstrate that *Cyp7b1⁻/⁻* spleens contain a normal compartment of pre-DCs and indicate that EBI2 is required in DCs after their differentiation to a CD4+ state. In further support of this conclusion, when sorted congenically distinguishable and violet tracer labeled pre-DCs from *Ebi2⁻/⁻* and *Ebi2⁺/⁻* mice were cotransferred into wild-type hosts, fewer *Ebi2⁻/⁻* CD4+ DCs were detectable 6 days later but these cells appeared to have divided a similar number of times as the control cells (*Figure 4G,H* and *Figure 4—figure supplement 1A*).

CD4+ DC development is strongly dependent on Notch signaling (*Caton et al., 2007*; *Lewis et al., 2011*). Flow cytometric analysis showed that the CD4+ DC remaining in *Ebi2⁻/⁻* mice had similar amounts of the Notch induced gene, ESAM, as in controls (*Figure 4—figure supplement 1B*). They also expressed similar amounts of the Notch target gene Deltex1 (Dtx1) (*Figure 4—figure*

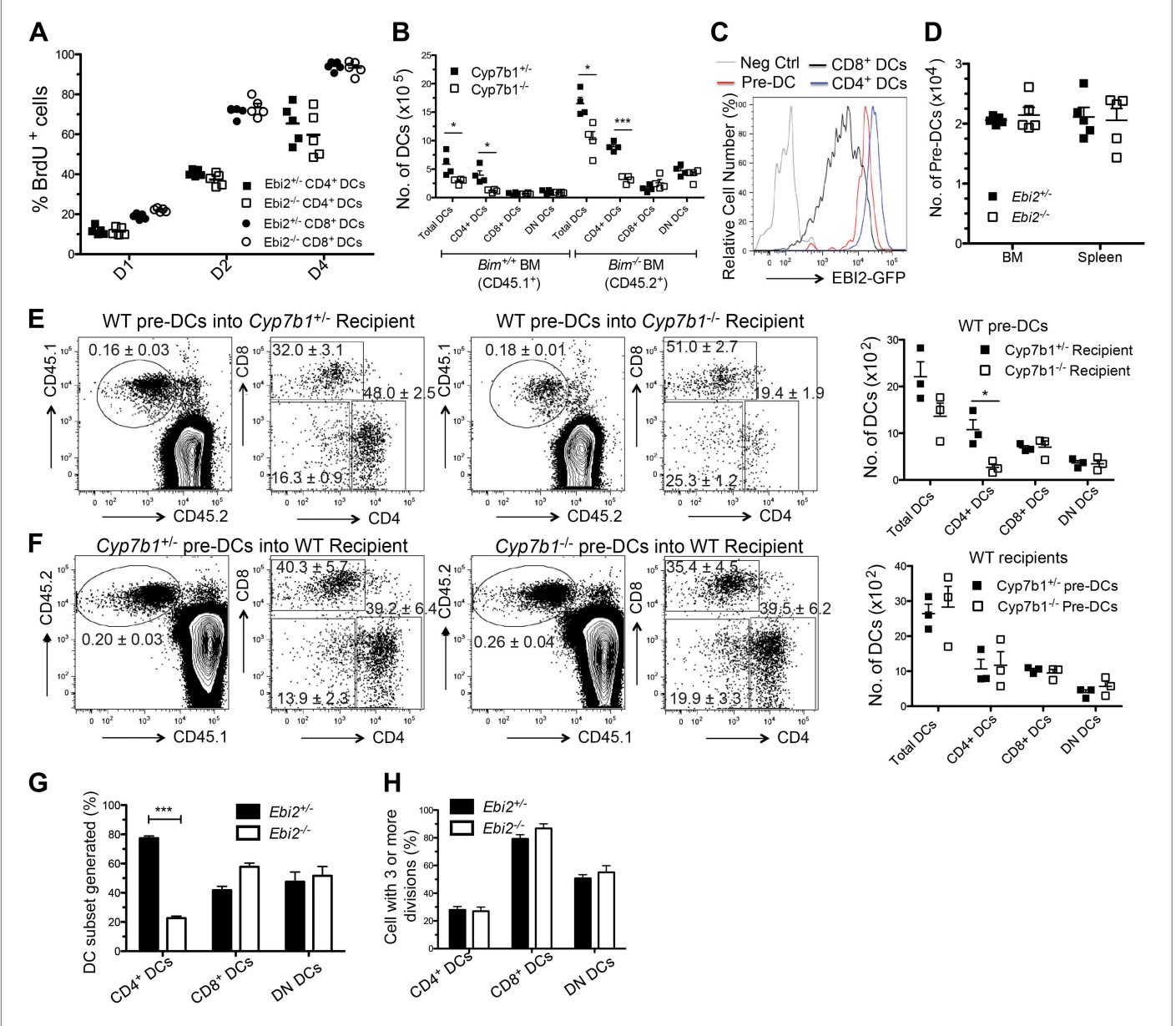

**Figure 4**. Normal mature DC turnover and pre-DC function in EBI2-deficient mice. (**A**) BrdU positive cells as percent of CD4+ and CD8+ DCs from *Ebi2+/−* or *Ebi2−/−* mice given BrdU in drinking water for 1, 2, or 4 days. Data are pooled from two to three experiments with five mice in each group. (**B**) *Cyp7b1+/−* or *Cyp7b1−/−* recipients were reconstituted with equal number of BM cells from *Bim−/−* (CD45.2) and *Bim+/+* (CD45.1) mice and DCs enumerated by flow cytometry (n = 4). (**C**) Flow cytometric detection of GFP fluorescence in gated splenic pre-DCs or DCs from *Ebi2GFP/+* or WT mice (negative control). Pre-DCs are defined as Lin− (CD19, B220, CD3, NK1.1, Ter119) CD11c+MHCII−CD172aintCD135+. (**D**) Number of pre-DCs from spleen and BM of *Ebi2+/−* and *Ebi2−/−* mice (n = 5 mice). (**E**) Sorted pre-DCs (1.0 × 10^5) from CD45.1+ WT mice were adoptively transferred into CD45.2+ *Cyp7b1+/−* or *Cyp7b1−/−* recipients. (**F**) Sorted pre-DCs (1.0 × 10^5) from CD45.2+ *Cyp7b1+/−* or *Cyp7b1−/−* mice were adoptively transferred into CD45.1+ WT recipients. 6 days after transfer, different subsets of CD45.1+(**E**) or CD45.2+ (**F**) pre-DC derived DCs were quantified by flow cytometric analysis, with the right plots in each pair being pre-gated as shown in each left plot (n=3 mice, one representative of two experiments). Graphs on right show a summary of data from three recipients of each type. (**G**) and (**H**) 1:1 Ratio mixed *Ebi2−/−* (CD45.2+) and *Ebi2+/−* (CD45.1+CD45.2+) pre-DCs were labeled with violet tracer and transferred into CD45.2+ WT recipients. 6 days after transfer, the appearance of DC subsets (**G**) and the frequency of each DC type that had divided three or more times (**H**) were analyzed by flow cytometry (n = 6). A description of the experimental scheme and an example of the flow cytometric data is included in the figure supplement.

The following figure supplements are available for figure 4:

**Figure supplement 1**. PreDC transfer strategy, Notch and IRF4-induced gene expression and effects of Flt3L on DC frequencies.

*supplement 1C*). IRF4 is another factor important for development of splenic CD4$^+$ DCs (*Tamura et al., 2005*) but EBI2-deficient CD4$^+$ DCs had normal levels of the IRF4 target Mmp12 (*Figure 4— figure supplement 1C*), suggesting that this transcription factor had been appropriately activated.

Flt3 ligand plays a role in the development and maturation of most splenic DC types (*Hashimoto et al., 2011*). To test if diminished access to Flt3 ligand might account for the reduction in CD4$^+$ DCs mice were inoculated with Flt3 ligand producing cells. This led to the expected expansion of both CD4$^+$ and CD8$^+$ DCs in control mice (*Figure 4—figure supplement 1D*). There was also expansion of DCs in the EBI2-deficient mice, but the selective deficiency in CD4$^+$ DCs remained (*Figure 4—figure supplement 1D*). Inoculation of GM-CSF producing tumor cells also expanded total CD11c$^+$ cell numbers but failed to restore the CD4$^+$ DC compartment in EBI2-deficient mice (not shown). These data suggest that insufficient access to Notch ligand, Flt3 ligand or GM-CSF cannot explain the reduced numbers of CD4$^+$ DCs.

## Restoration of CD4$^+$ DC in EBI2-deficient mice by LTβR agonism

Another factor that is important for homeostasis of splenic CD4$^+$ DCs is the cytokine LTα1β2 (*Kabashima et al., 2005*; *Wang et al., 2005*). In previous work we found that when mice lacked LTα1β2 not only were CD4$^+$ DCs reduced in number, but the level of LTβR on the cells was elevated, consistent with less ligand-mediated down-modulation (*Kabashima et al., 2005*). Flow cytometric analysis of cells from EBI2-deficient mice revealed elevation of LTβR on the remaining CD4$^+$ DCs while there was minimal change in the LTβR surface levels on CD8$^+$ DCs (*Figure 5A*). Consistent with the DC-deficiency being a consequence of reduced LTβR signaling, treatment for 6 days with an LTβR agonistic antibody (*De Trez et al., 2008*; *Stanley et al., 2011*) led to a significant rescue in the proportion of CD4$^+$ DCs present in the spleen (*Figure 5B,C*). In tissue sections, increased numbers of EBI2-deficient 33D1$^+$ DCs were evident in both the T zone and red pulp but not in the bridging channels (*Figure 5D*). A rescue in the proportion of CD4$^+$ DCs was also seen when *Cyp7b1$^{-/-}$* mice were reconstituted with BM from transgenic mice (*Ngo et al., 2001*) that overexpress LTα1β2 on B cells (*Figure 5E*). These data are consistent with the possibility that EBI2-mediated positioning of CD4$^+$ DCs between follicles and near LTα1β2-expressing B cells (*Ansel et al., 2000*) is required for maintenance of the CD4$^+$ DC compartment.

## Bridging channel positioning facilitates capture of systemic antigen

The EBI2-dependent positioning of many CD4$^+$ DCs in the MZ bridging channels, adjacent to the blood-rich MZ and red-pulp, suggests these cells will have enhanced access to particulate antigens compared to DCs situated more deeply in the T zone. To explore this idea we injected mice intravenously with sheep red blood cells (SRBCs) that had been labeled with the membrane dye, PKH26 (*Hagnerud et al., 2006*). 1 day later, flow cytometric analysis revealed marked capture of PKH26$^+$ antigen by splenic CD4$^+$ DCs whereas capture by CD8$^+$ DCs was considerably lower (*Figure 6A*). A similar extent of antigen capture by CD4$^+$ DCs was evident as early as 30 min after SRBC injection and PKH26$^+$ CD4$^+$ DCs were still evident in the spleen at 48 hr (*Figure 6—figure supplement 1A,B*). Analysis of tissue sections from mice 30 min after injection of PKH26-labeled SRBC and fluorescent CD11c antibody showed the expected high density of SRBC throughout the MZ (*Hagnerud et al., 2006*), including in regions overlapping with clusters of CD11c$^+$ DCs (*Figure 6—figure supplement 1C*). Control experiments indicated that the antigen capture detected by flow cytometry occurred in vivo and not during tissue preparation (*Figure 6—figure supplement 1D*). Comparison of the number of PKH26 antigen$^+$ DCs between wild-type and EBI2-deficient mice showed a marked deficiency in the latter (*Figure 6B*).

Although MZ bridging channel positioning situates CD4$^+$ DCs well for exposure to blood-borne antigens, this location may be less optimal for interaction with T cells. Previous work has shown that bridging channel DCs move into the T zone in response to LPS exposure (*De Becker et al., 2000*; *De Trez et al., 2005*) but it has been unclear whether this occurs upon immunization with other antigen types. Remarkably, SRBC injection led to very rapid movement of 33D1$^+$ DCs from bridging channels to the T zone, with a notable bias for positioning at the B-T zone interface (*Figure 6C*). The cells remained in this location for 1 day, but by 2 days the distribution of 33D1$^+$ DCs resembled more closely the unstimulated state (*Figure 6C*). Interestingly, the inward movement of 33D1$^+$ DCs following SRBC immunization occurred more rapidly than following injection with LPS and with a greater bias for the T-B boundary (*Figure 6—figure supplement 1E*). Confirming the importance of bridging

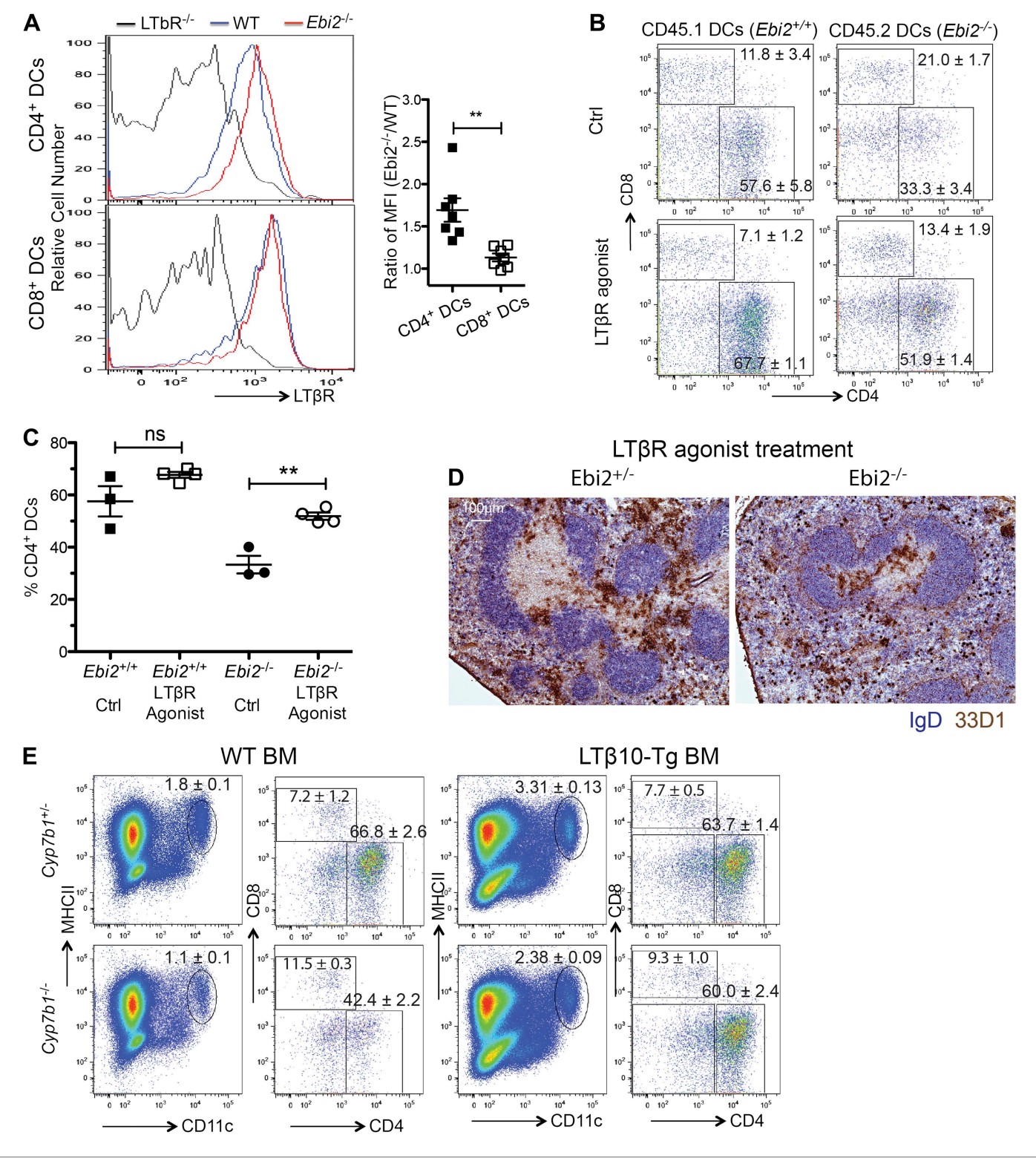

**Figure 5**. Rescue of EBI2-deficient CD4+ DCs by increased LTβR agonsim. (**A**) Surface staining of LTβR in indicated DC subsets from LTβR[−/−], WT, and Ebi2[−/−] mice (n=7 mice). Right panel shows the Ebi2[−/−]/WT LTβR staining median fluorescence intensity (MFI) ratio. (**B**) and (**C**) 1:1 Mixed Ebi2[−/−] (CD45.2) and Ebi2[+/+] chimeric mice were treated with LTβR agonist antibody or saline control. (**B**) Shows flow cytometric analysis for CD4 and CD8 on CD11c+ cells and (**C**) shows percentage of CD4+ DCs among total DCs from three to four mixed chimeras of each type. (**D**) Ebi2[+/−] or Ebi2[−/−] mice were treated with
Figure 5. Continued on next page

*Figure 5. Continued*

LTβR agonist antibody for 6 days. Immunohistochemistry staining for 33D1 and IgD in splenic sections from the indicated mice treated with LTβR agonist antibody. One representative picture of three replicated mice is shown. (**E**) *Cyp7b1$^{+/-}$* or *Cyp7b1$^{-/-}$* recipients were reconstituted with WT or lymphotoxin (LTβ10) transgenic (tg) BM cells and analyzed by flow cytometry for the indicated markers. Plots on right are pre-gated on MHCII$^+$CD11c$^+$ cells (n = 4 mice). *p<0.05, **p<0.01, ***p<0.001 by Student's T-test.

channel positioning in favoring access to blood-borne antigen, when mice were pretreated with unlabeled SRBCs for 6 hr to promote DC movement in to the T zone and then injected with PKH26$^+$ SRBC, the extent of PKH26$^+$ antigen capture by CD4$^+$ DCs was selectively diminished (*Figure 6D*).

To define the basis for the rapid repositioning of antigen-engaged DCs we tested whether the cells underwent changes in EBI2 or CCR7 expression. CCR7 was upregulated within 1.5 hr of SRBC injection (*Figure 6E*) and the cells responded more vigorously to the CCR7 ligands CCL21 and CCL19 than cells from saline injected mice (*Figure 6F*). By contrast, EBI2 expression and function was minimally affected following SRBC injection (*Figure 6F* and not shown). When mice were reconstituted with a mixture of CCR7-deficient and EBI2-deficient BM such that the majority of CD4$^+$ DCs were CCR7-deficient while most other cell types were CCR7 wild-type, injection of SRBC failed to cause DC mobilization from MZ bridging channels to the T-B boundary (*Figure 6—figure supplement 1F*), confirming the role of CCR7 in this repositioning event.

## Defective priming of T and B cell responses in EBI2-deficient mice

Finally we examined how the reduced CD4$^+$ DC frequency and defective DC positioning in EBI2-deficient mice affected CD4 T cell and T-dependent B cell responses. When EBI2-deficient mice that had received wild-type ovalbumin (Ova)-specific OTII TCR transgenic T cells were immunized with Ova-conjugated SRBCs there was a significant defect in the extent of T cell proliferation (*Figure 7A*). The responding T cells also showed less strong upregulation of ICOS and PD1 (*Figure 7A*). A similar defect in OTII T cell proliferation was observed when transfers were made into *Ebi2$^{-/-}$: Itgax-DTR* (CD11c-DTR) mixed BM chimeras that had been treated with diphtheria toxin (DT) such that most DCs remaining in the animals were EBI2-deficient whereas all hematopoietic CD11c$^-$ cell types were ~50% EBI2 wild-type (*Figure 7—figure supplement 1A*). This defect appeared to be selective to responses against particulate antigen as CD4 T cell proliferation following injection with soluble Ova or 33D1 antibody conjugated with Ova (*Dudziak et al., 2007*) was unaffected by EBI2-deficiency (*Figure 7—figure supplement 1B,C*).

To test for effects on induction of a T-dependent antibody response, EBI2-deficient or control mice were given transfers of wild-type HEL-specific Hy10 B cells and then immunized with HEL-SRBC. 3 days later the Hy10 B cells had undergone reduced proliferation in EBI2-deficient compared to control hosts (*Figure 7B*). A similar defect in Hy10 B cell proliferation was observed when transfers were made into DT-treated *Ebi2$^{-/-}$:Itgax-DTR* mixed BM chimeras (*Figure 7C*). When the Hy10 B cell response in EBI2-deficient hosts was examined at day 5, there was an ~14-fold reduction in total plasma cells and a ~7-fold reduction in germinal center B cells (*Figure 7D,E*). Analysis of isotype switched (IgG1 and IgG2b) plasma cell numbers showed they were more strongly affected (15- to 30-fold reduced) than IgM plasma cells (~7-fold reduced, *Figure 7D,E*). These findings provide strong evidence that EBI2 function in DCs is needed for mounting normal CD4 T cell-dependent B cell responses.

## Discussion

The high density of DCs in splenic MZ bridging channels has been appreciated for 30 years (*Witmer and Steinman, 1984*) yet the factors controlling this localization have been unknown. The above findings establish a crucial role for EBI2 and its oxysterol ligand, 7α,25-OHC, in positioning CD4$^+$ 33D1$^+$ DCs in MZ bridging channels and in LN interfollicular regions. EBI2 is also required for the maintenance of a CD4$^+$ 33D1$^+$ DC compartment of normal size. These roles of EBI2 in DCs are necessary for supporting normal CD4 T cell and B cell proliferative responses, and early plasma cell and germinal center responses to particulate blood-borne antigens. These observations are in close agreement with a study that appeared online at the time this work was submitted (*Gatto et al., 2013*). Our study additionally provides evidence that EBI2-mediated positioning of CD4$^+$ DCs in splenic MZ bridging

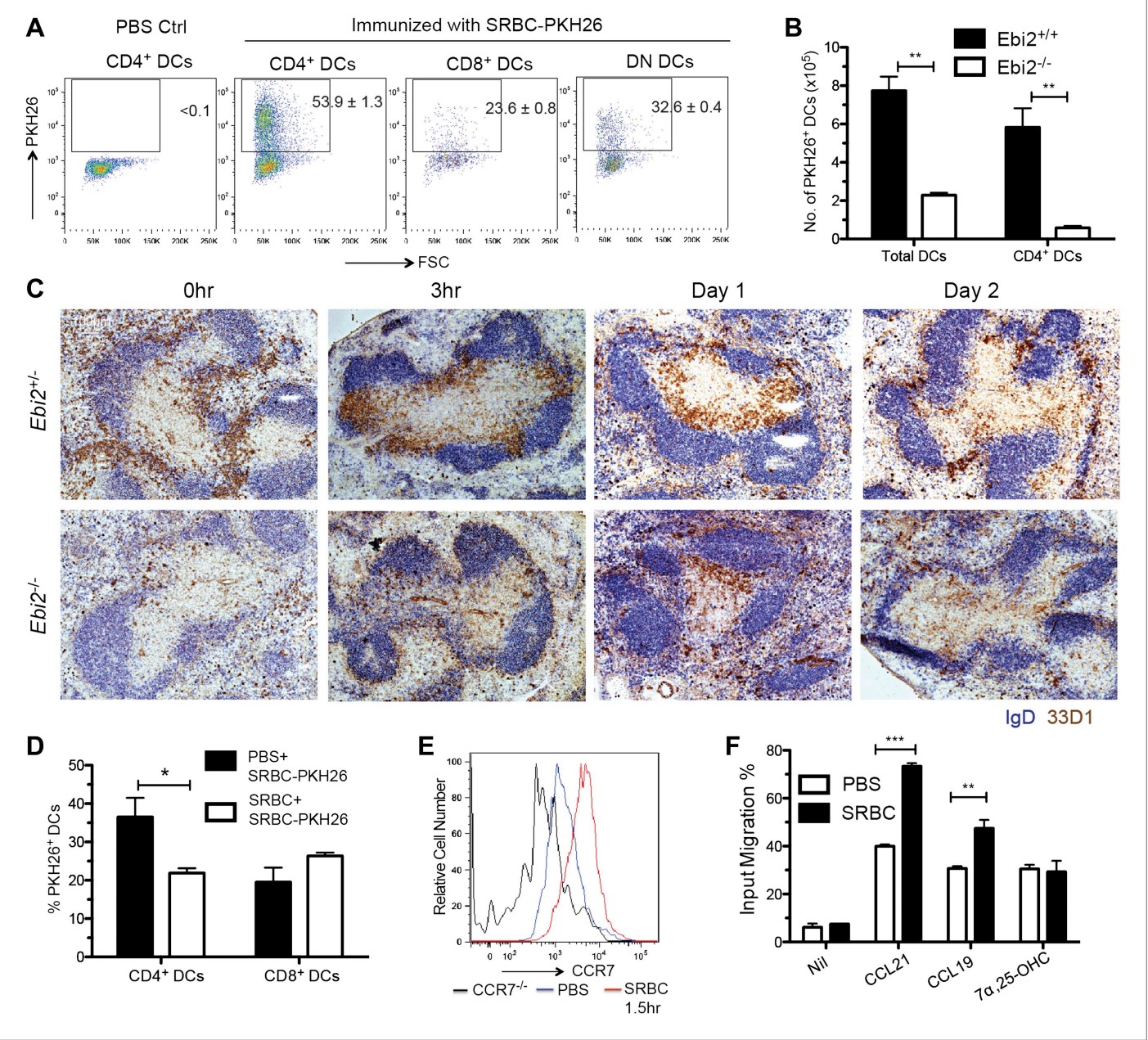

**Figure 6**. MZ bridging channel positioning facilitates DC capture of blood-borne particulate antigen and rapid movement to T zone. (**A**) Mice were i.v. immunized with SRBC-PKH26 or PBS control and analyzed 24 hr later by flow cytometry for PKH26 in gated DC subsets. Numbers indicate the frequency of cells within gate (mean ± SE, n = 4 mice). (**B**) $Ebi2^{+/+}$ or $Ebi2^{-/-}$ mice were immunized with SRBC-PKH26 and 24 hr later the number of PKH26+ total DCs or CD4+ DCs were quantified as in (**A**) (n = 4 mice of each type). (**C**) $Ebi2^{+/-}$ and $Ebi2^{-/-}$ mice were immunized with SRBCs and spleens isolated at the indicated time points and stained by immunohistochemistry for 33D1 and IgD. (**D**) Mice were pre-immunized with SRBCs or PBS and 6 hr later, immunized with SRBC-PKH26. 1 day later, the percentage of PKH26+ DCs was quantified by flow cytometry (n = 3 mice). (**E**) CCR7 surface staining of CD4+ DCs from $Ccr7^{-/-}$ mice or WT mice immunized with or without SRBC for 1.5 hr. One representative flow cytometric pattern of four mice is shown. (**F**) Migration of CD4+ DCs from PBS or SRBC immunized mice (1.5 hr) in response to medium only (nil), CCL19 (0.1 µg/ml), CCL21 (0.1 µg/ml), or 7α,25-OHC (0.5 nM). Bars show mean ± SE (n = 4 mice). *p<0.05, **p<0.01, ***p<0.001 by Student's T-test.

The following figure supplements are available for figure 6:

**Figure supplement 1**. Splenic distribution and DC capture of PKH26-labeled SRBC antigen and induction of DC repositioning by SRBC and LPS.

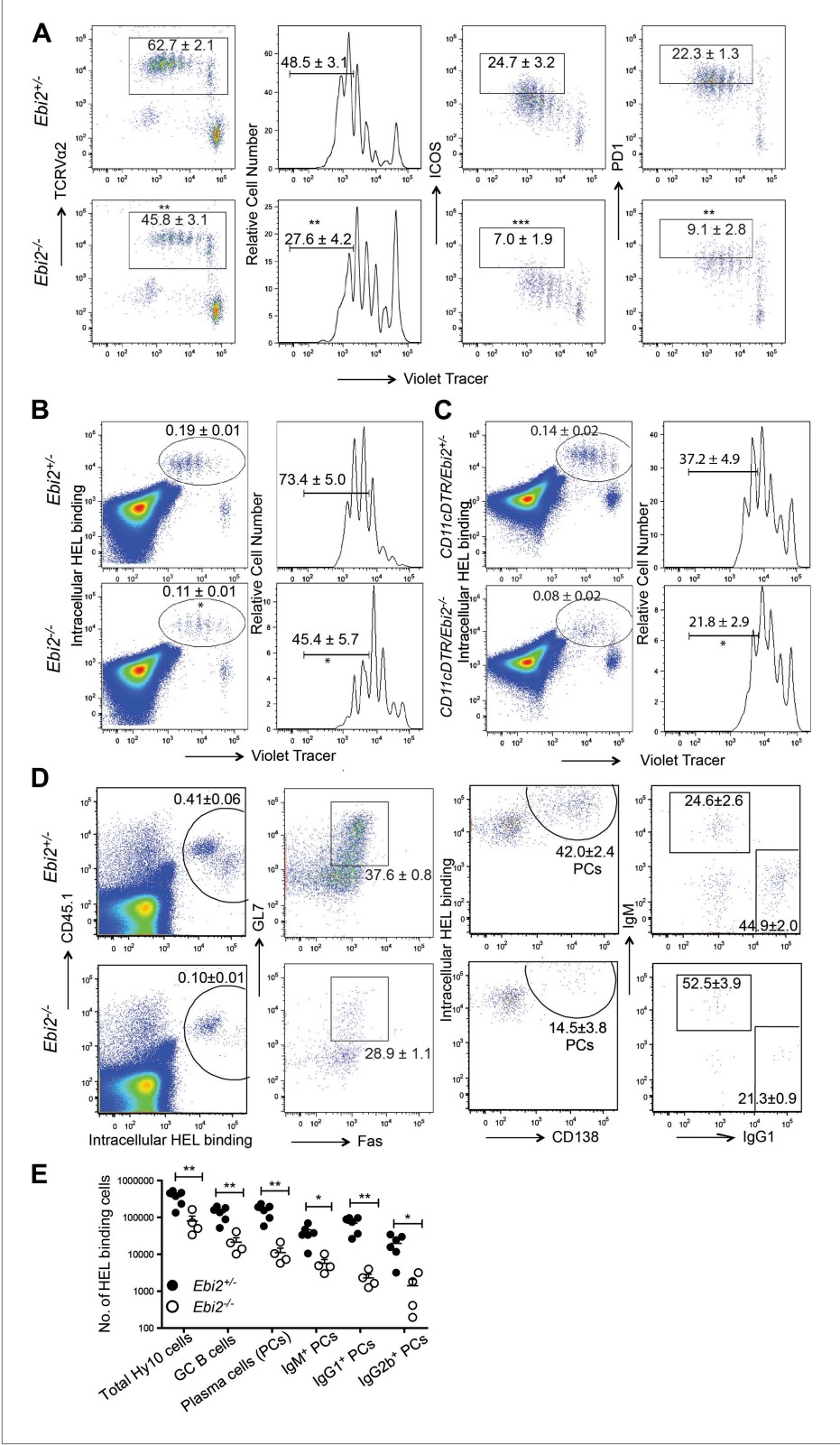

**Figure 7**. DC deficiency in *Ebi2^{–/–}* mice is associated with a reduced ability to support CD4 T cell and B cell responses to particulate antigen. (**A**) CD45.1^+ OTII splenocytes were labeled with cell trace violet and adoptively transferred into *Ebi2^{+/–}* and *Ebi2^{–/–}* mice. 1 day after transfer, mice were immunized with SRBC-OVA conjugate. 3 days post immunization, OTII T cell proliferation and expression of ICOS and PD1 was examined by flow cytometry. Left panels *Figure 7. Continued on next page*

*Figure 7. Continued*

are pre-gated on CD45.1⁺CD45.2⁻ cells and right three panels are further gated on TCR Vα2⁺ OTII cells. Numbers on gates indicate mean (±SE) frequencies for seven mice combined from two experiments. (**B**) and (**C**) Cell trace violet labeled Hy10 splenocytes were transferred into *Ebi2⁺/⁻* or *Ebi2⁻/⁻* recipients (**B**) or 1:1 mixed CD11cDTR and *Ebi2⁺/⁻* or *Ebi2⁻/⁻* BM chimeras (**C**). 1 day after cell transfer, mice were immunized with SRBC-HEL²ˣ and flow cytometric analysis of Hy10 B cell frequency and proliferation was conducted at day 3 after immunization. Hy10 B cells were identified as intracellular HEL-binding cells. For (**C**), mice were treated with DT at the day of Hy10 B cell transfer and again 2 days after cell transfer. Mean (±SE) cell frequency is shown next to each gate (n = 5–9 mice combined from two to three replicated experiments). (**D**) and (**E**) CD45.1⁺ Hy10 splenocytes were transferred into *Ebi2⁺/⁻* or *Ebi2⁻/⁻* recipients and mice were immunized with SRBC-HEL²ˣ. 5 days after immunization, HEL-binding Hy10 B cells and HEL-specific plasma cells of different Ig isotypes were quantified by flow cytometric analysis. Frequencies of GC B cells and total plasma cells are shown in plots (middle two panels) pre-gated on Hy10 B cells (gate shown in left panels). Frequencies of IgG1⁺ plasma cell are shown in plots (right panels) pre-gated on Hy10 plasma cells. Numbers on plots in D indicate mean (±SE) cell frequencies in the indicated gates and graph in (**E**) provides a summary of total HEL-binding cell numbers of the indicated types. Data are representative of three replicated experiments with four to six mice in each experiment. Each dot represents an individual mouse. *p<0.05, **p<0.01, ***p<0.001, by Student's T-test.

The following figure supplements are available for figure 7:

**Figure supplement 1**. EBI2 deficiency on DCs results in defective CD4+ T cell response to particulate antigen but not soluble and 33D1-coupled OVA.

channels promotes two important processes: (1) encounter with B cell-derived LTα1β2 that engages the DC LTβR, delivering a signal necessary for maintaining the homeostasis of the population; and, (2) efficient exposure to blood-borne particulate antigens and an ability to promptly access the T-B zone interface to stimulate T-dependent B cell responses (*Figure 8*).

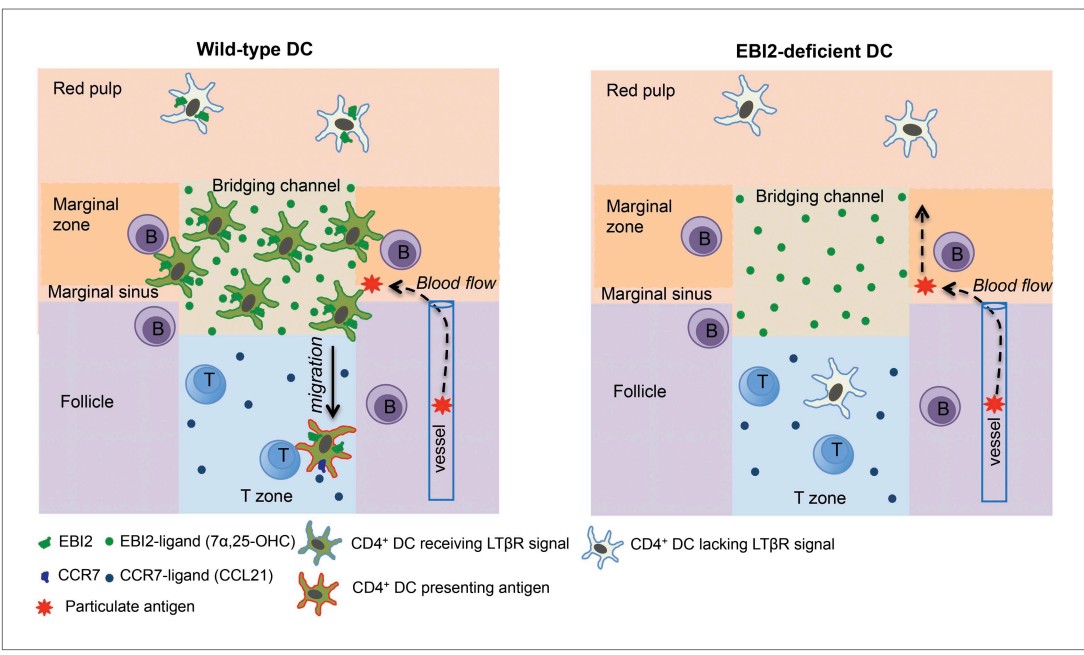

**Figure 8**. Model for the role of EBI2 in mediating marginal zone (MZ) bridging channel positioning of CD4⁺ DCs in the spleen. The major zones and cell types are labeled. EBI2 ligand is suggested to be produced in high amounts in the MZ bridging channel by stromal cells (not depicted) and acts to promote positioning of EBI2ʰⁱCD4⁺ DCs (that are also 33D1⁺) in this region. This in turn juxtaposes them to MZ and follicular B cells, resulting in exposure to B cell-derived LTα1β2 (not depicted). LTβR engagement on the DC transmits a homeostasis-promoting signal. This positioning also exposes the cells to particulate antigens traveling with blood flow through the marginal sinus and the MZ. Following particulate antigen exposure, DCs migrate in a CCR7-dependent manner from the bridging channel in to the T zone where they present antigen to helper T cells.

Our finding of a similar disruption in CD4$^+$ DC numbers and bridging channel positioning in mice incapable of making 7α,25-OHC (Ch25h- and Cyp7b1-deficient) and in mice unable to properly degrade 7α,25-OHC (Hsd3b7-deficient) (*Russell, 2003*; *Yi et al., 2012*) provides strong evidence that the critical function of EBI2 in CD4$^+$ DCs is to mediate their correct positioning. The alternative possibility that EBI2 is providing a maintenance signal that is independent of its positioning role seems unlikely as *Hsd3b7*$^{-/-}$ mice with elevated 7α,25-OHC would not be expected to have the same defect in EBI2 signaling as mice that lack 7α,25-OHC. Cyp7b1 and Ch25h are abundantly expressed in the outer follicle and in fibroblastic reticular cells (*Yi et al., 2012*) and our present findings suggest the necessary source of 7α,25-OHC for bridging channel positioning and maintenance of CD4$^+$ DCs is stromal. Further experiments will be needed to determine the nature of the stromal cells expressing these enzymes in MZ bridging channels and LN interfollicular regions.

In previous work that established a role for LTα1β2 in maintaining normal numbers of CD4$^+$ DCs within the spleen, a slight reduction in the rate of *Ltbr*$^{-/-}$ DC turnover was observed (*Kabashima et al., 2005*; *Wang et al., 2005*). The basis for our inability to observe a statistically significant reduction in DC turnover in EBI2-deficient mice is not clear but may indicate that the cells continue to receive low level LTβR engagement. Precisely how a reduction in LTβR engagement leads to a decrease in CD4$^+$ DC numbers needs further investigation. The sufficiency of increased EBI2 expression to lead to greater numbers of CD4$^+$ DCs may be a consequence of increased LTβR signaling since increased LTβR engagement is adequate to increase CD4 DC numbers (*Kabashima et al., 2005*; *Wang et al., 2005*). To account for our inability to observe changes in the CD4$^+$ DC turnover rate or rescuing effects of antagonizing apoptosis when EBI2 function was lacking, we suggest that CD4$^+$ DCs are lost in EBI2-deficient mice at a similar rate irrespective of how long it has been since they were generated from pre-DCs or last proliferated. It seems possible that the cells suffer from the combined effect of mispositioning due to EBI2-deficiency and lack of expression of molecules downstream of LTβR signaling that control interactions with other cell types, with the outcome that the CD4$^+$ DCs are more frequently engulfed by phagocytes or become caught in blood flow and lost from the spleen. In this regard, it is notable that mice lacking CD47 or its partner protein SIRPα, have a similar deficiency in splenic CD4$^+$ DCs (*Hagnerud et al., 2006*; *Van et al., 2006*; *Saito et al., 2010*). Although we did not observe alterations in CD47 or SIRPα expression by DCs in EBI2-deficient mice (T. Yi and J. Cyster, unpublished observation), it is possible that the function of these molecules in mediating cell–cell interactions or in regulating engulfment by phagocytic cells is somehow altered in EBI2-deficient DCs.

The marked defect in T cell proliferative responses in EBI2-deficient mice following injection with antigen-conjugated to SRBCs, but intact response to soluble Ova, observed here and in the recent study of Brink and coworkers (*Gatto et al., 2013*), suggests that EBI2 function in splenic DCs is most important for responses against particulate antigens. We provide evidence that one reason for this differing dependence on EBI2 is that particulate antigen is not able to freely access DCs already positioned within the T zone. This finding is consistent with other studies showing an inverse relationship between the size of molecules and their ability to diffusively access the white pup (*Nolte et al., 2003*). The intact response of EBI2-deficient mice to 33D1-Ova was unexpected but suggests that the defective response to SRBC-antigen is not solely a consequence of reduced CD4$^+$ DC numbers. Instead, these data suggest that appropriate positioning in MZ bridging channels, and thus in close proximity to the marginal sinus and blood-rich MZ, is critical for supporting the response against particulate antigens. This requirement might reflect both the improved efficiency with which DCs in this region can capture blood-borne particles and the ability to be triggered for rapid movement into the T zone (*Figure 8*). Although 33D1 coupling ensured efficient delivery of antigen to CD4$^+$ DCs it did not promote their movement into the T zone (*Chappell et al., 2012* and data not shown).

The reduction in CD4 T cell responses in mice lacking EBI2$^+$ DCs described here and by *Gatto et al. (2013)* might be sufficient to account for the reduced plasma cell and germinal center responses observed in both studies. However, a recent report showed that when antigen was targeted to 33D1$^+$ DCs as an antibody conjugate, it promoted some B cell activation events in a manner that did not depend on T cell help (*Chappell et al., 2012*). A number of studies have shown that DCs can directly augment B cell responses (*MacPherson et al., 1999*; *Balazs et al., 2002*; *Jego et al., 2005*; *Qi et al., 2006*). Our finding that 33D1$^+$ DCs move to the B-T zone interface following particulate antigen immunization also seems consistent with the possibility that these DCs interact with activated

B cells as well as T cells. Future studies will be needed to determine whether EBI2-dependent DC–B cell interactions contribute to driving early B cell activation events during responses to particulate antigens.

The surface markers ICOS and PD1 are highly expressed by T follicular helper (Tfh) cells (*Vinuesa and Cyster, 2011*). The less effective induction of ICOS and PD1 on T cells responding to SRBC-Ova in EBI2-deficient hosts suggests that antigen presentation by CD4$^+$ DCs may be important in favoring induction of an early B-helper phenotype in CD4 T cells. In addition to efficiently presenting antigen in the context of MHC class II (*Pooley et al., 2001*; *Dudziak et al., 2007*) it is possible that appropriately activated CD4$^+$ DCs bias T cell differentiation in a manner that favors provision of B cell help. Consistent with early induction of Tfh cell properties in activated T cells augmenting B cell responses, when T cells lack the 'master regulator' of Tfh cell differentiation, Bcl6, they are poorly able to support extrafollicular plasmablast responses (*Lee et al., 2011*). The reduced induction of Tfh-phenotype cells likely also contributes to the reduced germinal center response.

Our findings indicate that CD4$^+$ DC positioning is controlled by a balance of EBI2 and CCR7 expression and ligand responsiveness. Under homeostatic conditions, EBI2 has a dominant influence and promotes positioning in MZ bridging channels. However, following activation by antigen exposure the balance shifts in favor of CCR7 and the cells move promptly into the T zone (*Figure 8*). While we suggest that direct exposure to antigen triggers DC movement, we do not exclude the possibility that CCR7 upregulation is promoted indirectly as a consequence of cytokine production by other antigen-exposed cells. The basis for the antigen activated DCs favoring the B-T zone interface compared to the central T zone is not yet clear but this does not seem to depend on EBI2 expression. These data add to a series of findings showing how differential responsiveness to chemoattractants emanating from adjacent zones determines cell position (*Reif et al., 2002*; *Cyster, 2005*; *Bromley et al., 2008*). Although LPS causes CCR7 upregulation on splenic DCs and promotes their movement to the T zone (*De Becker et al., 2000*; *De Trez et al., 2005*; *Seul et al., 2012*) we found that LPS-mediated repositioning happened more slowly and with a more uniform distribution of DCs through the T zone than occurred following SRBC immunization. The DC sensor triggered following SRBC immunization is not clear but in preliminary experiments we do not find a requirement for Myd88 (unpublished observation). SRBCs rapidly become opsonized by complement and presumably also by natural antibody and it is possible that these factors contribute to SRBC capture by CD4$^+$ DCs and to triggering CCR7 upregulation. It seems possible that detection of damaged endogenous RBCs may be a trigger for DC maturation as a number of important pathogens, most notably the malaria parasite, propagate inside RBCs, insert foreign proteins in their membranes and alter their functional properties.

## Materials and methods

### Mice

C57BL/6NCr and C57BL/6-cBrd/cBrd/Cr (CD45.1) mice at age 7–9 weeks were from National Cancer Institute (Frederick, MD). *Ebi2*$^{-/-}$ mice (*Pereira et al., 2009*) were backcrossed to C57BL/6J for eleven generations. These mice carry an *eGFP* gene inserted in place of the *Ebi2* open reading frame. *Ch25h*$^{-/-}$ mice (*Bauman et al., 2009*) were backcrossed 10 generations to C57BL/6, *Cyp7b1*$^{-/-}$ mice (*Rose et al., 2001*) were backcrossed to C57BL/6J for 6 generations. *Hsd3b7*$^{-/-}$ mice were backcrossed to C57BL6/J for two generations and maintained on chow containing 0.5% cholic acid and pan-vitamin supplemented water (*Shea et al., 2007*). HEL-specific Hy10 mice, OVA-specific OTII TCR-transgenic mice, *Itgax-DTR* transgenic mice and LTβ line 10 (LTβ10)-transgenic mice were on a C57BL/6J background. For BrdU labeling, mice were given BrdU containing (0.5 mg/ml) drinking water ad lib. For bone marrow chimeras, mice were lethally irradiated by exposure to 1300 rads of g-irradiation in two doses 3 hr apart and bone marrow cells (2–5 × 10$^6$) were transferred through the tail vein. Chimeric mice were analyzed 6–10 weeks after reconstitution. Animals were housed in a specific pathogen-free environment in the Laboratory Animal Research Center at the University of California, San Francisco, and all experiments conformed to ethical principles and guidelines approved by the Institutional Animal Care and Use Committee.

### Cell adoptive transfer and immunizations

For antibody responses, 1 × 10$^5$ Hy10 B cells, OT II T cells, or TCR7 T cells were adoptively transferred into *Ebi2*$^{-/-}$ or matched control recipients. 1 day after cell transfer, recipients were i.p. immunized with 2 × 10$^8$ SRBCs conjugated with low affinity HEL$^{2x}$ or OVA. 33D1-OVA is produced by transfecting 293T

cells with 33D1-OVA plasmid (kindly provided by Michel Nussenzweig) and further purified through protein G column. To visualize cell proliferation, cells were labeled with CellTrace violet cell proliferation kit (Invitrogen; Grand Island, NY). For LTβR agonist treatment, mice were treated with agonistic anti-LTβR antibody (3C8, provided by Carl F Ware) by i.p. injection of 100 µg of antibody every 3 days for 6 days.

Sheep blood was obtained from Colorado Serum Company (Denver, CO). For conjugation of SRBCs with PKH26 (Sigma-Aldrich; St Louis, MO), the procedure was conducted following the manual instruction with 25 µM PKH26 per 250 µl SRBC alsevers. For conjugation of SRBC-HEL$^{2\times}$, SRBCs were first washed with PBS for three times, mixed with HEL$^{2\times}$ (10 µg/ml SRBC alsevers), crosslinked with EDCI (1-ethyl-3-(3-dimethylaminopropyl) carbodiimide, Sigma-Aldrich) for 30 min, and washed three times to remove the free HEL. For SRBC-OVA conjugation, OVA protein (Sigma-Aldrich) was first cross-linked with HEL with glutaraldehyde and further conjugated to SRBC in a same manner as SRBC-HEL$^{2\times}$ conjugation.

## Flow cytometry and cell sorting

For EBI2 surface staining, cells were incubated with a goat anti-EBI2 (A20) polyclonal antibody (Santa Cruz Biotechnology; Dallas, TX), then a biotinlynated anti-goat antibody, and streptdavidin Alex647 (Jackson Immunoresearch; West Grove, PA). Antibodies against CD11c, DCIR2 (33D1), DEC205, CD4, CD8α, MHCII, CD172α, CD24, CD135, Siglet H, CD45.1, CD45.2 were obtained from Biolegend (San Diego, CA) or eBioscience (San Diego, CA). For intracellular plasma cell staining, spleens were perfused and digested with RPMI1640 containing 2% fetal bovine serum, collagenase type 4 (0.25 mg/ml), and DNase I (20 µg/ml) for 45 min in a 37°C incubator. Immediately after digestion, enzymes were inactivated by adding 2 mM EDTA and splenocytes were isolated through a 70-µm cell strainer. Cells were first stained with the fixable viability dye eFluor450 (eBioscience), blocked with anti-FcR for 10 min (UCSF Cell Culture Core), and stained with anti-CD45.1 (Biolegend), anti-CD45.2 (Biolegend), anti-B220 (Biolegend), anti-CD138 (BD Biosciences; San Jose, CA) for 20 min on ice. The cells were then washed twice and fixed with a BD Cytofix/Cytoperm fixation kit (BD Biosciences). Fixed cells were stained with Alexa647-conjugated HEL and anti-Ig isotype on ice for 15 min. For DC sorting, spleens were digested with 1.6 mg type II collagenase (Worthington Biochemical; Lakewood, NJ) and DNase I for 30 min at 37°C. Digested spleens were meshed into single cell suspension through a 100-µm cell strainer in PBS buffers containing 2% FCS and 2 mM EDTA. DCs were pre-enriched with anti-CD11c microbeads (Miltenyi Biotec) and further sorted on a FACSAira III with a 100-µm nozzle to purities of over 99%. For pre-DC sorting, Type II collagenase and DNase I digested splenocytes were stained for anti-CD11c Biotin and anti-Biotin microbeads (Miltenyi Biotec; Auburn, CA). Cells were pre-enriched with LS-column and further sorted on Moflow to purities of over 95%. For pre-DCs transfer experiments, 1.0 × 10$^5$ sorted pre-DCs (define as Lin$^-$CD11c$^+$CD135$^+$ Sirpα$^{int}$MHCII$^-$) were adoptively transferred into non-irradiated congenic 3- to 4-week-old mice. 6 days after transfer, enriched CD11c cells from the whole mouse spleen was analyzed by flow cytometry with 0.7~3.0 × 10$^6$ events collected.

## Immunohistochemical staining

Cryosections of 7 µm were fixed and stained as described (*Pereira et al., 2009*) with: FITC-conjugated anti-IgD (11-26c.2a; BD Biosciences), biotin conjugated DEC205 (Biolegend), bio-conjugated 33D1 (Biolegend). For staining of 33D1, a tyramide kit was used (TSA Biotin System; Perkin Elmer; Waltham, MA). Images were captured with a Zeiss AxioOberver Z1 inverted microscope.

## Cell line transfection and BM retroviral transduction

For EBI2 transduction of BM-derived cells, BM cells were harvested 4 days after 5-flurouracil (Sigma) injection and cultured in the presence of recombinant IL-3, IL-6, and mouse stem cell factor (SCF) (100 ng/ml; Peprotech; Rocky Hill, NJ). BM cells were spin-infected twice with a retroviral construct expressing EBI2 or truncated nerve growth factor receptor (NGFR) and an IRES–truncated human CD4 cassette as a reporter. 1 day after the last spin infection, hCD4 positive cells were FACS sorted and injected into lethally irradiated C57BL/6 recipients. Around 50% of the hematopoietic cells in recipients were hCD4 positive 6 weeks after reconstitution.

## Oxysterol, Chemokine, and chemotaxis assay

7, 25-OHC was synthesized as previously described (*Hannedouche et al., 2011*). Mouse recombinant CCL19 and CCL21 were obtained from R&D systems. The chemotaxis assay was conducted

as previously described (*Yi et al., 2012*) and 5 µM transwell plates were obtained from Corning Life Sciences.

## RNA isolation and real-time RT-PCR

Total RNA was isolated from ~2.0 × 10$^4$ double-sorted cells with the RNEasy kit (Qiagen; Hilden, Germany) and in-column DNA digestion. Real-time PCR was performed using SYBR Green PCR Mix (Roche; Mannheim, Germany) and an ABI prism 7300 sequence detection system (Applied Biosystems; Foster City, CA). *Hprt* mRNA levels were used as internal controls. Sequences for PCR primers were as previously described (*Yi et al., 2012*).

## Acknowledgements

We thank Robert Brink for providing HEL$^{2x}$, Carl Ware for LTβR agonist antibody, D Russell, J Gustafsson and R Lathe for mice, and Tina Christakos, Jinping An and Ying Xu for excellent technical assistance.

## Additional information

### Funding

| Funder | Grant reference number | Author |
|---|---|---|
| Howard Hughes Medical Institute | | Jason G Cyster |
| National Institutes of Health | AI40098 | Jason G Cyster |
| Cancer Research Institute Postdoctoral fellowship | | Tangsheng Yi |

The funders had no role in study design, data collection and interpretation, or the decision to submit the work for publication.

### Author contributions

TY, Research conception and design, acquisition of data, analysis and interpretation of data, and manuscript preparation; JGC, Research conception and design, interpretation of data, and manuscript preparation

### Ethics

Animal experimentation: All animal experiments were performed in accordance with the Guide for the Care and Use of Laboratory Animals of the National Institutes of Health. Experimental protocols were reviewed and approved by the Institutional Animal Care and Use Committee (AN 087331-02A).

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
