## [Decision Letter]

Thank you for sending your work entitled “EBI2-mediated bridging channel positioning supports splenic dendritic cell homeostasis and particulate antigen capture” for consideration at *eLife*. Your article has been favorably evaluated by a Senior editor and 2 reviewers, one of whom is a member of our Board of Reviewing Editors.

The following individuals responsible for the peer review of your submission want to reveal their identity: Michel Nussenzweig (Reviewing editor).

The Reviewing editor and the other reviewer discussed their comments before we reached this decision, and the Reviewing editor has assembled the following comments to help you prepare a revised submission.

Our major concern was that while the positioning and survival phenotypes are extremely clear, the analysis of the immune response presented in Figure 7 seems rather limited, especially given the outstanding track record of the Cyster lab in analyzing B cell responses. The main issue is that these experiments rely exclusively on proliferation of transferred monoclonal cells. We would like to see some evidence that EBI2 expression in DCs affects antibody production/GC formation/affinity maturation in a polyclonal response. Perhaps using CD11c-DTR mixed chimeras could provide such a system?

---

## [Author Response]

*Our major concern was that while the positioning and survival phenotypes are extremely clear, the analysis of the immune response presented in Figure 7 seems rather limited, especially given the outstanding track record of the Cyster lab in analyzing B cell responses. The main issue is that these experiments rely exclusively on proliferation of transferred monoclonal cells. We would like to see some evidence that EBI2 expression in DCs affects antibody production/GC formation/affinity maturation in a polyclonal response. Perhaps using CD11c-DTR mixed chimeras could provide such a system*?

We thank the reviewers for their careful assessment of our study. In connection with this concern, we have added data to the figure 7 supplement (Figure 7–figure supplement 1) showing the OTII T cell response in DT-treated *Ebi2*^*-/-*^*:CD11c-DTR* mixed BM chimeras, supporting the conclusion that the reduced wild-type T cell response is due to the role of EBI2 in CD11c^+^ cells, most likely DCs. We have also added data to figure 7 panels D and E showing that the wild-type GC B cell response in the Hy10 B cell transfer system is reduced in *Ebi2*^*-/-*^ hosts.

We did attempt an experiment of the type suggested by the reviewers, where CD11c- DTR:EBI2 KO (or WT) mixed BM chimeras were treated with DT (at days -1, 1 and 3), immunized with SRBC (d0) and then analyzed for the frequency of polyclonal plasma cells and GC cells at day 5. However, we found that this approach was problematic as the DT treatment caused almost complete ablation of plasma cells and GC B cells derived from the CD11c-DTR BM (Author response image 1A). This effect is consistent with past and very recent findings that CD11c-DTR (and CD11c-YFP) is expressed in plasmablasts and GC B cells (Hebel et al., 2006 Eur J Immunol 36, 2912; Racine et al., 2008 J Immunol 181, 1375; Baumjohann et al., 2013 Immunity 38, 596). While this approach did not allow us to examine the impact of selective EBI2-deficiency in DCs on WT B-lineage cells, the experiment did allow assessment of the response of EBI2- deficient B cells in mice that lack EBI2 on CD11c^+^ DCs while retaining EBI2 on approximately half of all non-CD11c expressing cells (such as helper T cells). This analysis showed the plasma cell response was reduced 3-fold and the GC response about 2.5-fold compared to the response in control CD11c-DTR:WT mixed BM chimeras (Author response image 1B). This reduction was greater than observed for EBI2- deficient B cells in SRBC immunized Ebi2^-/-^:WT mixed BM chimeras, where the plasma cell response was reduced only 25% and the GC response was not reduced (Author response image 1C). The greater defect in the DT treated Ebi2^-/-^:CD11c-DTR chimeras is consistent with a role for EBI2^+^ DC in supporting events necessary for the polyclonal anti-SRBC plasma cell and GC responses.Author response image 1**(A, B)** Mixed (1:1) CD11cDTR (CD45.1^+^) and Ebi2^-/-^ or Ebi2^+/+^ (CD45.1ʱCD45.2^+^) BM chimeric mice were immunized with SRBCs and flow cytometric analysis of B cell responses performed five days later. DT was administrated day -1, day 1, and day 3 after immunization. **(A)** Left panel shows a representative example of the CD45.1 and CD45.2 staining profile of total splenocytes. Right panels show representative flow cytometric pattern of gated CD45.1^+^CD45.2^+^ cells and CD45.1^+^ for plasma cells (defined as B220^low^CD138^+^) and germinal center cells (defined as  GL7^+^Fas^+^). **(B)** Representative flow cytometric pattern of gated CD45.1^+^CD45.2^+^ Ebi2^+/+^ (upper) or Ebi2^-/-^ (lower) cells. Mean (±SE) frequency of plasma cells and germinal center B cells are indicated next to each gate (n=3 mice). **(C)** Mixed Ebi2^+/+^ (CD45.1+) and Ebi2^-/-^ (CD45.1^+^CD45.2^+^) BM chimeric mice were immunized with SRBCs and flow cytometric analysis was performed on B cell responses five days after immunization. Mean (±SE) frequency of plasma cells and germinal center B cells (GL7^+^B220^+^) are indicated next to each gate (n=4 mice, representative of 3 experiments).

However, given the complexity of interpreting this experiment, we felt that these data would distract from the clarity of the experiments using transgenic B cell transfers and we have therefore not included them in the manuscript. Also in relation to this point, we now cite the recent report of Gatto et al., (Nat. Immunol. 2013, advanced online), that includes data showing the endogenous GC response to SRBC is reduced in mice conditionally deficient for EBI2 in CD11c-Cre expressing cells. Since EBI2 is minimally expressed in GC B cells, their data are most consistent with the reduced polyclonal GC B cell response being due to an EBI2 requirement in CD11c^+^ DCs.

Given that many recent studies of B cell biology take advantage of Ig-transgenic (or knockin equivalent) mice for their entire analysis, we do not feel that our use of the approach as one part of the present study is out of line with currently accepted practice in the field. It will be valuable in the future to perform studies with mouse models that allow more selective ablation of DCs, such as the recently developed Zbtb46-DTR mice, but such studies would take many months to perform. We hope that the reviewers will agree that further addressing this concern, while valuable in the longer term, is not essential in the context of the current submission.